# A Radiomic Approach for Evaluating Intra-Subgroup Heterogeneity in SHH and Group 4 Pediatric Medulloblastoma: A Preliminary Multi-Institutional Study

**DOI:** 10.3390/cancers16122248

**Published:** 2024-06-18

**Authors:** Marwa Ismail, Hyemin Um, Ralph Salloum, Fauzia Hollnagel, Raheel Ahmed, Peter de Blank, Pallavi Tiwari

**Affiliations:** 1Department of Radiology, University of Wisconsin-Madison, Madison, WI 53706, USAptiwari9@wisc.edu (P.T.); 2Nationwide Children’s Hospital, Columbus, OH 43205, USA; 3Department of Medicine, University of Wisconsin-Madison, Madison, WI 53792, USA; 4Department of Neurological Surgery, University of Wisconsin-Madison, Madison, WI 53792, USA; 5Department of Pediatrics, Cincinnati Children’s Hospital Medical Center, Cincinnati, OH 45229, USA; 6Departments of Medical Physics and Biomedical Engineering, University of Wisconsin-Madison, Madison, WI 53792, USA

**Keywords:** medulloblastoma, SHH, Group 4, texture, shape, risk stratification

## Abstract

**Simple Summary:**

Medulloblastoma (MB) is the most common malignant brain tumor in children and has a dismal prognosis. A challenge with MB is identifying patients who could be candidates for reduced doses of radiation therapy, but are still treated effectively, as well as those that need intensified doses. Recently, MB was classified into four molecular subgroups with distinct clinical outcomes (WNT, SHH, Group 3, and Group 4). Though two of these subgroups (SHH and Group 4) are known for their intermediate prognosis, wide disparities of outcomes have been reported within each of these subgroups. This work aims to develop a prognostic signature using radiomics (computationally derived tumor measurements), acquired on MRI scans, to risk-stratify patients within the SHH and Group 4 subgroups. Our signature includes two key attributes that capture aspects of the disease microenvironment. We believe that our signature will provide a better understanding of the disease’s heterogeneity and, hence, develop better personalized treatment plans.

**Abstract:**

Medulloblastoma (MB) is the most frequent malignant brain tumor in children with extensive heterogeneity that results in varied clinical outcomes. Recently, MB was categorized into four molecular subgroups, WNT, SHH, Group 3, and Group 4. While SHH and Group 4 are known for their intermediate prognosis, studies have reported wide disparities in patient outcomes within these subgroups. This study aims to create a radiomic prognostic signature, medulloblastoma radiomics risk (mRRisk), to identify the risk levels within the SHH and Group 4 subgroups, individually, for reliable risk stratification. Our hypothesis is that this signature can comprehensively capture tumor characteristics that enable the accurate identification of the risk level. In total, 70 MB studies (48 Group 4, and 22 SHH) were retrospectively curated from three institutions. For each subgroup, 232 hand-crafted features that capture the entropy, surface changes, and contour characteristics of the tumor were extracted. Features were concatenated and fed into regression models for risk stratification. Contrasted with Chang stratification that did not yield any significant differences within subgroups, significant differences were observed between two risk groups in Group 4 (*p* = 0.04, Concordance Index (CI) = 0.82) on the cystic core and non-enhancing tumor, and SHH (*p* = 0.03, CI = 0.74) on the enhancing tumor. Our results indicate that radiomics may serve as a prognostic tool for refining MB risk stratification, towards improved patient care.

## 1. Introduction

One of the biggest challenges in pediatric medulloblastoma (MB), the most frequent malignant brain tumor in children, is the accurate risk stratification of patients, which serves as a key determinant in accurate treatment pathways. MB accounts for 20% of all pediatric intracranial tumors and its overall survival remains inadequate with a five-year survival rate of around 70–75% [1]. When relapse occurs, it is nearly always fatal, making the proper selection of upfront chemotherapy and craniospinal irradiation (CSI) based on risk stratification even more important [2]. The current clinical risk stratification approach is Chang’s classification, which classifies patients into standard/average- and high-risk, using age and clinical parameters such as the extent of the resection and the presence of metastases [3,4]. Recent efforts in molecular profiling and gene expression resulted in categorizing MB patients into four unique subgroups (WNT, SHH, Group 3, and Group 4) [5], which have been used in conjunction with Chang’s classification in clinical trials for risk-adapted treatments [5]. Unfortunately, with molecular classification, wide disparities among patients have been reported, revealing intra-subgroup heterogeneity for some of these subgroups [6,7]. For instance, recent studies reported a dismal prognosis in patients of the SHH group with the TP53 mutation (five-year overall survival of 41% ± 9%), but more favorable outcomes were observed in younger patients of the same subgroup with TP53 wild-type tumors (five-year survival rate of 81% ± 5%) [7,8,9]. Similarly, while Group 4 MB patients are considered to generally have an intermediate prognosis, studies have demonstrated that there is a wide variation in patient outcomes within this subgroup [8,9,10]. In the absence of approaches that can predict the patients’ outcomes precisely and resolve the extensive heterogeneity that MB tumors are known for, the need is underscored for complementary risk stratification tools that can resolve the intra-subgroup heterogeneity and identify the different risk levels within each subgroup. This may provide further insights in personalized treatment plans (therapy intensification/de-escalation) beyond and complementary to molecular profiling or Chang’s stratification.

Radiomics has recently emerged as a powerful tool to quantitatively analyze medical images by extracting feature attributes that capture the unique cues of the tumor micro-environment, reflecting the hallmarks of the tumor biology [11]. Of note, there has been extensive work carried out that explored radiomics in adult brain tumors for several clinical problems, including diagnosis, prognosis, and predicting treatment response [12,13,14]. However, these approaches have only been recently exploited in pediatric brain tumor research, with the primary focus being utilizing clinical predictors or gene expression profiling for pediatric tumors [15]. Recent works in the literature exploited traditional radiomic features (textural and morphological) for survival analysis and identifying molecular subtypes in MB [1,16,17,18,19,20,21,22,23,24,25,26,27,28,29]. For instance, the studies in [16,17,18,19,20] attempted to employ statistical techniques, such as univariate and multivariate logistic regression, to predict survival in MB. Regression models (e.g., LASSO) have been incorporated in these works, followed by a Kaplan–Meier estimate for risk stratification. Similarly, the studies in [1,20,21,22,23,24,25,26] employed several machine-learning classifiers (e.g., SVM), along with some statistical techniques, such as logistic regression and ANOVA, to predict the four molecular subgroups in MB. While the aforementioned approaches investigated some of the clinical challenges with MB, there is a lack of approaches that utilize radiomics to predict the different outcomes within the same molecular subgroup, notably SHH and Group 4.

This work presents a radiomic descriptor, “medulloblastoma radiomics risk” (mRRisk), to capture the intra-subgroup heterogeneity within the subgroups that are known for wide disparities in outcomes, namely, Group 4 and SHH patients. Our goal is to provide a non-invasive prognostic tool that can capture the tumor micro-environment and its substantial biological heterogeneity on imaging. This can enable us to provide more robust, personalized prognostic insights. mRRisk will include image features pertaining to (1) *3D textural and entropy features* to capture micro-architectural differences within the tumor confines which could reflect the disorderly nature and high heterogeneity of aggressive tumors, and (2) *3D surface-based topological features* extracted from the tumors which provide cues regarding surface irregularities and lesion aggressiveness. In addition, our pipeline uniquely accounts for the developing anatomy of pediatric brains, by registering the subject brains to age-appropriate atlases. Our rationale in this work is that mRRisk features, encompassing the intra- and peri-tumoral confines, can provide a comprehensive analysis of the tumor heterogeneity and aggressiveness, teasing out the differences between low- and higher-risk patients. This can enable further stratification within the individual molecular subgroups and, hence, allow for personalized treatment regimens with targeted therapies.

## 2. Materials and Methods

### 2.1. Overview

Figure 1 illustrates the pipeline of our proposed work. Following pre-processing and tumor segmentation, we extract radiomic features that quantify both the morphological and textural attributes of the pediatric MB tumors. Following feature extraction, our prognostic signature, mRRisk, is constructed by concatenating the two feature families into one vector. Statistical methods were then applied on mRRisk, particularly, multivariate logistic regression (including Elastic Net, LASSO regression, and ridge regression) for feature pruning and selecting the features that are statistically significant in risk stratification while shrinking those that do not contribute to risk assessment. The top features were then employed to create a continuous survival risk score that stratifies all the patients into low- and high-risk groups, within each molecular subgroup (SHH and Group 4).

### 2.2. Data Curation

Our analysis was conducted on a total of 70 pediatric MB subjects (48 in Group 4 and 22 in SHH molecular subgroup). Studies were retrospectively collected from three independent sites: Site 1: Cincinnati Children’s Hospital Medical Center (CCHMC) (*n* = 22, Group 4 (*n* = 14), SHH (*n* = 8)), Site 2: Children’s Hospital Los Angeles (CHLA) (*n* = 31, Group 4 (*n* = 18), SHH (*n* = 13)), and Site 3: Children’s Hospital of Philadelphia (*n* = 17, Group 4 (*n* = 16), SHH (*n* = 1)). The scans were performed from 2000 up to the date of IRB approved data (5/16/2019). Scans were acquired with 1.5 T and 3 T MRI Philips (Ingenia, Achieva) and Siemens scanners. The inclusion criteria used for our datasets were: (a) availability of Gd-T1w axial-view MRI scans; (b) patients with only MB tumors; (c) known molecular status, Chang’s classification status, and overall survival; and (d) acceptable diagnostic quality of the MRI scans, as identified by the collaborating radiologists. Details of patient demographics and MRI acquisition information are listed in Table 1.

### 2.3. Pre-Processing and Feature Extraction

All the Gd-T1w images were bias-corrected to remove the scan inhomogeneities [30]. Then, the ground truth annotations for all the tumors of the pediatric MB scans were generated via consensus across two experienced board-certified neuro-radiologists (Expert 1 with nine years of experience, and Expert 2 with eight years of radiology experience) using 3D Slicer [31]. Each MB tumor was segmented into the enhancing tumor region, the edema region, and the non-enhancing tumor region + cystic core. This was followed by registering the scans to age-appropriate atlases to account for the changing anatomical structures during the different developmental stages [32]. Finally, intensity standardization was conducted using the approach described in [33]. 

Feature extraction then followed, which involved 214 texture features as well as 18 morphological features that were extracted from the different tumor regions as well as the tumor habitat that encompasses all the tumor regions. Namely, the texture features [34] encompass gradient, Haralick, Laws, Gabor, and COLLAGE (gradient entropy) features [35] and are computed for every voxel of every tumor region. We utilize these per-voxel measurements to compute first-order statistics (mean, median, standard deviation, skewness, and kurtosis) per feature for every tumor region. This resulted in 1070 textural features for every tumor region, as well as for the tumor habitat. Additionally, 18 morphological features were extracted from the different tumor regions. Namely, four local features that capture surface-based irregularities (Curvedness, Sharpness, Shape Index, and Total Curvature) were computed from each region. The four local features were computed from the constructed isosurfaces of the tumor regions, followed by computing the first and second fundamental forms of the surfaces [36]. Gaussian and mean curvatures were then computed from the fundamental forms per voxel. Finally, the local features were derived from the Gaussian and mean curvatures. In addition, 14 morphological features capturing the global contour characteristics of the tumors were computed from each tumor region and the tumor habitat. Extracted features are based on an Insight Segmentation and Registration Toolkit (ITK) implementation (www.itk.org). The features are volume, major axis length, minor axis length, eccentricity, elongation, orientation, perimeter, roundness, equivalent spherical radius, equivalent spherical diameter, flatness, elongation shape factor, compactness, and integrated intensity. Similar to the scheme for the textural features, five statistics were calculated for each of the four extracted surface-based features, and then concatenated with the 14 global features, resulting in a 34×1 vector, per each tumor region as well as the tumor habitat. All the extracted features were then concatenated to construct mRRisk for every patient.

### 2.4. Regression Analysis

All the extracted feature attributes were then employed within logistic regression analysis models with α representing the regularization parameter. Specifically, Least Absolute Shrinkage and Selection Operator (LASSO) (L1-regularization), α=1; ridge regression (L2 regularization), α=0; and elastic net (combining the penalty terms of both LASSO and ridge regression (0<α<1)) were employed within the regression models to conduct survival risk assessment [37]. We utilized the models to conduct feature selection, and then create a continuous survival risk score for each subject. Based on the fitted risk models, a threshold was identified to risk-stratify patients into low- and high-risk groups [38], within SHH and Group 4 molecular subgroups, individually. Log-rank test along with Kaplan–Meier (KM) survival analysis were then performed to see how the survival rate varies between the two identified risk groups. Performance metrics were computed to assess the efficacy of our survival prognostication models, such as hazard ratios (HRs), risk of experiencing the event of interest at a time point [39], 95% Confidence Interval (CI), level of uncertainty about the point estimates [40], and Concordance Index (C-index), a measure of the probability of concordance between the predicted and the observed survival [41]. All the computations were conducted in-house using RStudio (V.4.3.1).

We compared the performance of our prognostic models to risk-stratify MB patients within Group 4 and SHH molecular subgroups, with the following strategies: (1) shape features alone, (2) texture features alone, and (3) Chang’s stratification.

## 3. Results

Our analysis was conducted using three different data combinations to assess the robustness of our approach and the resilience of the extracted feature sets. Specifically, data from each site was used once for testing while combining the data from the other two sites for training. 

### 3.1. Risk-Stratifying MB Patients in Group 4 Subgroup

#### 3.1.1. Employing Shape Features Alone for Risk Stratification

When employing shape features on Group 4 subgroup subjects, significant differences were observed between the non-enhancing tumor + cystic core subcompartments of the subjects, resulting in two risk groups (Figure 2a,b). The differences were observed when using Site 1 as a test set (*p* = 0.0035, C-index = 0.5), using LASSO regression, as well as when using Site 3 as a test set (*p* = 0.025, C-index = 0.74), using Elastic Net model (alpha = 0.5). The top features selected using our regression models across the different experiments included the perimeter, elongation, and minor axis length, as well as some surface features, such as the median of curvedness, skewness of sharpness, and kurtosis of shape index.

#### 3.1.2. Employing Texture Features Alone for Risk Stratification

When employing texture features on Group 4 subgroup subjects, significant differences were observed between the tumor habitat, the edema, and the non-enhancing tumor + cystic core subcompartments of the subjects, resulting in two risk groups (Figure 2c,d). The differences were observed on (a) the tumor habitat when using Site 3 as a test set (*p* = 0.0017, C-index = 0.69), using ridge regression; (b) the tumor habitat using Site 2 as a test set (*p* = 0.04, C-index = 0.5), using the Elastic Net model (alpha = 0.38); (c) the edema subcompartment using Site 2 as a test set (*p* = 0.05, C-index = 0.58), using ridge regression; and (d) the non-enhancing tumor + cystic core subcompartment using Site 1 for testing (*p* = 0.04, C-index = 0.82), using LASSO regression. The top features selected using our regression model across the different experiments included the skewness of Laws features and median of Collage feature (information measure of correlation).

#### 3.1.3. Employing mRRisk Signature for Risk Stratification

Interestingly, when combining both feature families (shape and texture) into the mRRisk signature to risk-stratify the Group 4 subgroup, the C-indices for the risk stratification results improved. For instance, when using Site 3 for testing, significant differences between the two risk groups were observed on the tumor habitat (C-index = 0.7 vs. 0.69 when using texture features alone, while the *p*-values were the same (0.0017) for both experiments (Figure 3)). Similarly, when using Site 2 for testing, the tumor habitat and edema exhibited significant differences across the two risk groups with improved C-indices when using mRRisk (0.52, 0.6 vs. 0.5, 0.58 when using texture features alone), but the *p*-values were similar for both experiments (*p* = 0.04 for habitat and 0.05 for edema). Figure 4 shows heatmaps that illustrate the qualitative differences between the two risk groups identified within the Group 4 subgroup using our radiomic features. 

### 3.2. Risk-Stratifying MB Patients in SHH Subgroup

#### 3.2.1. Employing Shape Features Alone for Risk Stratification

When employing shape features on SHH subgroup subjects, significant differences were observed between the non-enhancing tumor + cystic core subcompartments as well as the tumor habitat of the subjects, resulting in two risk groups (Figure 5a,b). The differences were observed on the tumor habitat when using Site 2 as a test set (*p* = 0.04, CI = 0.7), using ridge regression, as well as on the non-enhancing tumor + cystic core subcompartments when using Site 1 as a test set (*p* = 0.01, CI = 0.8), using ridge regression. The top features selected using our regression model included the perimeter, roundness, minor and major axes lengths, and compactness, as well as some surface features, such as the median of sharpness, kurtosis of total curvature, variance of curvedness, and median of shape index. 

#### 3.2.2. Employing Texture Features Alone for Risk Stratification

When employing texture features on SHH subgroup subjects, significant differences were observed between the enhancing tumor subcompartment of the subjects, resulting in two risk groups (Figure 5c). The differences were observed when using Site 1 as a test set (*p* = 0.03, CI = 0.74), using the Elastic Net model (alpha = 0.54). The top features selected using our regression model included the Laws and Collage features with its different types of statistics. Figure 6 shows heatmaps that illustrate the qualitative differences between the two risk groups identified within the SHH subgroup using our radiomic features. 

Interestingly, similar to the Group 4 subgroup experiments, the results improved when combining the texture and shape features into mRRisk. For instance, when using Site 1 as a test set, significant differences were observed on the enhancing tumor between the two risk groups, with C-index = 0.8 compared to 0.74 using texture features alone (Figure 5d). 

Since the SHH subgroup is known to have wide disparities in outcomes that are associated with age [42], with younger patients having better survival outcomes, we attempted to see if there were any age-wise significant differences across our two identified risk groups using mRRisk. Interestingly, the subjects that exhibited significant differences in risk levels when employing textural features of the enhancing tumor also exhibited significant differences in age (*p* = 0.09) [43,44].

### 3.3. Risk-Stratifying MB Patients in SHH and Group 4 Subgroup Using Chang’s Stratification

When employing the current clinical classification criteria (Chang’s classification) for risk stratification within the two subgroups (SHH and Group 4), differences across the survival risk categories were not observed to be significant between the subjects from the same subgroup. For instance, in the Group 4 subgroup, performance metrics of *p* = 0.1 and CI = 0.5 were obtained when using Site 3 for testing, *p* = 0.47 and CI = 0.5 when using Site 2 for testing, and *p* = 0.54, CI = 0.5 when using Site 1 for testing. Similarly, significant differences were not observed when employing Chang’s classification to risk-stratify patients within the SHH subgroup (*p* = 0.7, CI = 0.6 when using Site 1 for testing).

We show the feature importance graphs for all the conducted experiments in the Appendix A. In Appendix A, we show the feature importance graphs for mRRisk descriptor features for both SHH and Group 4 subgroups, whereas the y-axis represents the feature, and the x-axis represents the F-score for each feature. Similarly, Appendix A illustrates the feature importance graphs for shape features for both SHH and Group 4 subgroups. Finally, Appendix A illustrates the feature importance graphs for texture features for both SHH and Group 4 subgroups. 

## 4. Discussion

In this study, we presented a radiomic prognostic signature, “medulloblastoma radiomics risk” (mRRisk), that risk-stratifies medulloblastoma (MB) patients within the individual molecular subgroups, namely, the SHH and Group 4 subgroups. mRRisk combines hand-crafted textural and morphological features that capture the tumor heterogeneity and disorderly nature within its confines, and, hence, may offer additional insights to resolve the intra-subgroup heterogeneity of MB tumors. MB is widely recognized as having four molecular subgroups with correlated clinical outcomes and prognosis; however, it is also reported that there is a wide disparity of outcomes within the individual subgroups [7,9,45]. This underscores the need for quantifying the tumor heterogeneity, which can help identify patients, within the same subgroup, with low risk that can benefit from de-escalated therapy from those with high risk and in need of intensified treatment strategies. While there are many radiomic approaches in the literature that aimed to carry out molecular subgroup classification [46], our study uniquely seeks to utilize those radiomic tools to further sub-stratify patients within the individual subgroups. Further optimization of mRRisk with rigorous validation on large multi-institutional cohorts would allow for an enriched risk stratification. Specifically, our prognostic signature can help reduce the long-term toxicities within the children with MB that are recognized as average-risk by providing additional prognostic insights to tailor their treatment intensity. Additionally, the signature can also enable the identification of patients that are true candidates for therapy intensification. This can lead to incorporating the signature in clinical trials that aim to carry out therapy de-escalation as well as therapy intensification, which could improve patients’ outcomes and treatment planning.

Our preliminary results showed that morphological radiomic attributes that capture the surface-based irregularities of the tumor, as well as its global contour characteristics, yielded two distinct survival risk groups within both the SHH and Group 4 molecular subgroups. Specifically, significant differences were observed across the non-enhancing tumor + cystic core compartments in Group 4 patients (*p* = 0.0035 on the test set) (Figure 2a). Interestingly, the same top features emerged when using different sites for testing (sites 1 and 3), individually, yielding significant results, indicating the robustness of our radiomic features. Similarly, the morphological attributes identified two risk levels within SHH group patients when employing those features on the tumor habitat (*p* = 0.04 on the test set) (Figure 5a), and on the non-enhancing tumor + cystic core compartments (*p* = 0.019 on the test set) (Figure 5b). Employing textural features also allowed for the identification of two statistically significant risk groups for both the SHH and Group 4 subgroup. For instance, entropy-based features capturing the frequency content in localized regions (e.g., Gabor) and features capturing the degrees of match of the voxel neighborhoods (e.g., Laws) helped sub-stratify patients from both subgroups when employed on the tumor subcompartments (e.g., *p* = 0.0017 on the tumor habitat of Group 4 (Figure 2c), and *p* = 0.03 on the enhancing tumor of SHH subgroup) (Figure 5c). Interestingly, combining the texture and morphological feature families into our prognostic signature, mRRisk, improved the performance metrics, specifically, the Concordance Index, obtained for risk stratification across the different test sets. These results suggest that combining the curvature local changes on the tumor surface with global contour attributes, in addition to the textural and gradient entropy changes, may provide surrogate quantitative attributes to quantify tumor heterogeneity, offering additional insights into risk assessment. Our approach may provide complimentary biomarkers that aid towards a more reliable risk assessment in pediatric MB.

There are many works in the literature that have attempted to employ radiomics in either predicting outcomes [16,17,18,19,20,28,29] or classifying molecular subgroups [1,21,22,23,24,25,26,27] for MB patients. However, there is a dearth in radiomic approaches that attempted to delve deeper into the molecular characteristics for MB and enable risk stratification within those subgroups, which would allow for the interpretation of the wide disparities among those patients. Very few studies attempted to sub-stratify patients within the individual molecular subgroups and utilized molecular profiling and other clinical approaches for this purpose. For instance, in a study by Schwalbe et al. [15], the authors attempted to quantify the substantial heterogeneity within each molecular subgroup in MB, by conducting molecular profiling, including DNA methylation analysis. Interestingly, the authors were able to identify seven subgroups within the four subgroups, two for SHH (stratified based on age), two for Group 3 (high- and low-risk), and two for Group 4 (high- and low-risk), while the WNT subgroup remained unchanged. Interestingly, when we attempted to assess if there were any significant differences in age between our two risk-stratified groups within the SHH subgroup [7,15,45], significant differences in age (*p* = 0.09) were observed between the two risk-stratified groups, for one of our conducted experiments. We could not identify any other age-wise significant differences within our performed experiments on the SHH subgroup, which is likely due to the small sample size of our dataset of patients with SHH (*n* = 22). 

Our study did have limitations. First, while multi-institutional, our sample size is limited for this study, with 48 subjects in Group 4 and 22 in SHH. The limited sample size partly limited the statistical power of the study and affected our results, where significant differences could not be observed on all three test sets in the different data combinations, for the two subgroups. Pediatric MB is a rare disease, so the curation of large multi-institutional studies is often challenging. Our goal is to continue curating multi-institutional MB studies with known molecular subgroup information so we can validate our approaches on larger external cohorts. We are also working on curating data from clinical trials (e.g., ACNS0331), where subjects were all uniformly treated and had known molecular subgroup characteristics. This will allow us to conduct a survival analysis on larger cohorts to possibly sub-stratify patients with similar molecular characteristics. Secondly, we did not have all MRI modalities available for this analysis (i.e., T2w and FLAIR) due to either the unavailability of the scan, or because of the poor quality of the scans leading to their exclusion from our analysis. Our current efforts with data curation consider the scans with all three available sequences.

## 5. Conclusions

This study presents a radiomic prognostic signature to risk-stratify medulloblastoma patients within the individual molecular subgroups. With the reported disparities in outcomes within the molecular subgroups and the substantial heterogeneity in medulloblastoma, our approach attempts to address the need for additional tools, besides clinical approaches, to quantify those differences, towards a more reliable risk assessment. Our results show promise in radiomic tools to identify different risk levels within patients that share the same molecular characteristics.

## Figures and Tables

**Figure 1 cancers-16-02248-f001:**
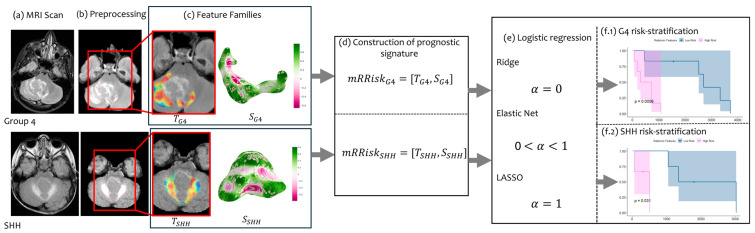
Pipeline of the proposed framework for risk stratification of SHH and Group 4 subgroups. T, S stand for texture features and shape features, respectively.

**Figure 2 cancers-16-02248-f002:**
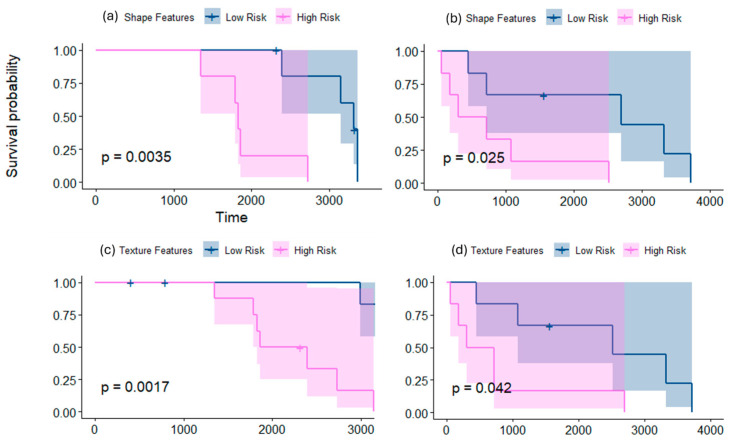
Kaplan–Meier curves for the statistically significant results when survival analysis is conducted using shape (**a**,**b**) and texture (**c**,**d**) features, individually, for risk stratification within Group 4 molecular subgroup. (**a**,**b**) show the KM curves when employing shape features on the non-enhancing tumor + cystic core subcompartments on datasets 3 and 1 as test sets, respectively. (**c**,**d**) show the KM curves when employing texture features on the tumor habitat of dataset 3 and the non-enhancing tumor + cystic core of dataset 1, as test sets, respectively. The *x*-axis represents the survival time in days, whereas *y*-axis represents the survival probability.

**Figure 3 cancers-16-02248-f003:**
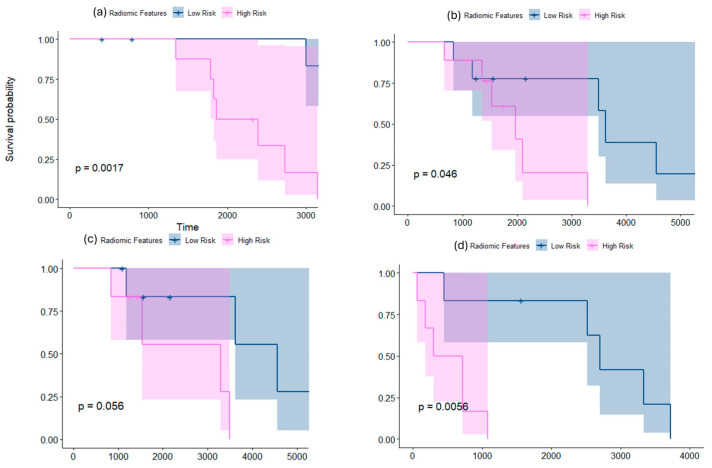
Kaplan–Meier curves for the statistically significant results when survival analysis was conducted using mRRisk signature, combined, for risk stratification within Group 4 molecular subgroup. (**a**) shows the KM curve when employing the features on the tumor habitat on dataset 3 as the test set. (**b**,**c**) show the KM curves when employing the features on the tumor habitat and the edema of dataset 2, respectively. (**d**) shows the results when employing the features on the non-enhancing tumor + cystic core of dataset 1 as a test set.

**Figure 4 cancers-16-02248-f004:**
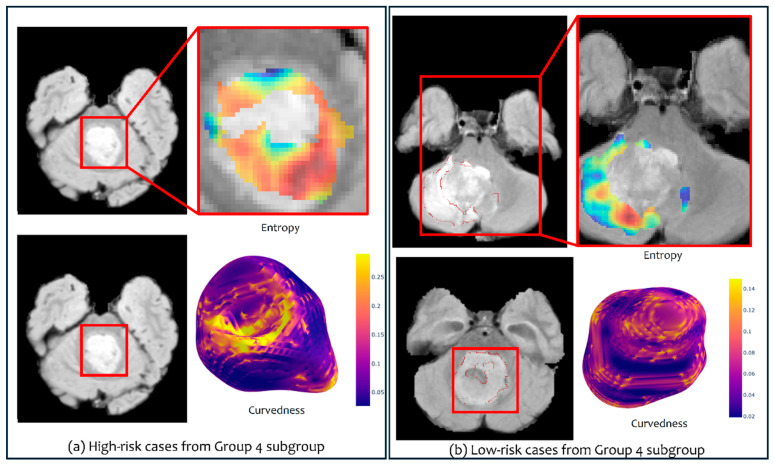
Heatmaps illustrating texture and shape features for high- (**a**) and low-risk (**b**) cases from Group 4 subgroup. The heatmaps shown are for the Collage (entropy) feature and the curvedness feature.

**Figure 5 cancers-16-02248-f005:**
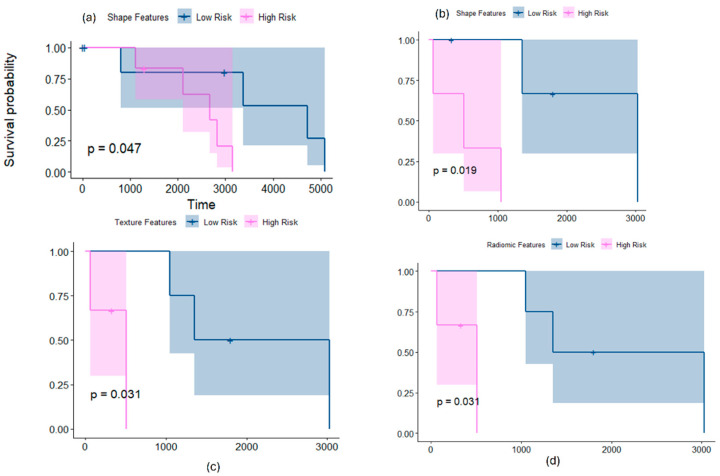
Kaplan–Meier curves for the statistically significant results when survival analysis is conducted using shape (**a**,**b**) and texture (**c**) features, individually, and (**d**) mRRisk, within SHH molecular subgroup. (**a**,**b**) show the KM curves when employing shape features on the tumor habitat and the non-enhancing tumor + cystic core subcompartments on datasets 2 and 1 as test sets, respectively. (**c**) shows the KM curves when employing texture features on the enhancing tumor of dataset 1 as a test set, and (**d**) shows the curves when employing mRRisk on the enhancing tumor of dataset 1 as a test set. The *x*-axis represents the survival time in days, whereas *y*-axis represents the survival probability.

**Figure 6 cancers-16-02248-f006:**
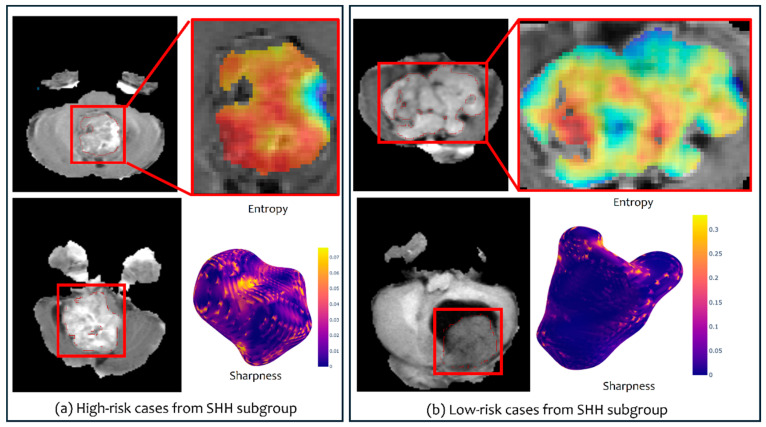
Heatmaps illustrating texture and shape features for high- (**a**) and low-risk (**b**) cases from the SHH subgroup. The heatmaps shown are for the Collage (entropy) feature and the sharpness feature.

**Table 1 cancers-16-02248-t001:** Patient demographics and MRI acquisition information across our multi-institutional data.

Site		CCHMC		CHLA		CHOP	
N		G4 N = 14	SHH N = 8	G4 N = 18	SHH N = 13	G4 N = 16	SHH N = 1
Age, mean (SD)		7.4 (3.8)	9.6 (6.6)	8.1 (4)	3.4 (3.3)	10.7 (3.45)	8.9
Sex	Male	12 (85.7)	5 (62.5%)	13 (72.2%)	5 (38.4%)	14 (87.5%)	
	Female	2 (14.3)	3 (37.5%)	5 (27.8%)	8 (61.6%)	2 (12.5%)	1 (100%)
Scan type		T1-FFE axial post-contrast		T1-FFE axial post-contrast		T1-FFE axial post-contrast	
MR acquisition type		2D		2D		2D	
Scanning sequence		Gradient-recalled		Gradient-recalled		Spin-echo	
Sequence variant		Steady-state		Steady-state		Segmented k-space/Spoiled/Oversampling phase	
Pixel spacing (mm)		0.46–1		0.46–1		2	
Slice thickness (mm)		Mean = 5.4 mm		Mean = 5.4 mm		Mean = 5.4 mm	

## Data Availability

The MRI scans obtained from CCHMC and CHLA are protected through institutional compliance at the local institutions. The clinical repository of these patient scans can be shared per specific institutional review board (IRB) requirements. Upon reasonable request, a data-sharing agreement can be initiated between the interested parties and the clinical institution following institution-specific guidelines. Data from CHOP were obtained from the Children’s Brain Tumor Network (CBTN), based on an established agreement between the senior author and CBTN. We will release the segmentations obtained from the CBTN studies into the CBTN network for future research purposes. CBTN membership can be obtained following the guidelines provided on their website to obtain access to the scans.

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
