# Peer review of "A Radiomic Approach for Evaluating Intra-Subgroup Heterogeneity in SHH and Group 4 Pediatric Medulloblastoma: A Preliminary Multi-Institutional Study"

_cancers, 2024, doi:10.3390/cancers16122248_

Round 1

Reviewer 1 Report

Comments and Suggestions for Authors

The article presents a study aimed at developing a radiomic prognostic signature, called mRRisc, to identify risk levels within the SHH and Group 4 subgroups of pediatric medulloblastoma (MB). This study involves the analysis of MRI scans from 70 MB subjects from three institutions. By extracting and analyzing 232 radiomic features, the authors aim to improve risk stratification and ultimately enable more personalized treatment plans for patients within these subgroups.

In my opinion, the article is well-written and addresses a critical challenge in pediatric oncology: the accurate risk stratification of medulloblastoma patients. Radiomics certainly has the potential to complement existing molecular and clinical stratification methods.

From a statistical standpoint, the study is robust: the use of advanced regression techniques such as LASSO, Ridge Regression, and Elastic Net is a significant strength.

I have a few considerations, primarily related to the complexity of the subject matter. Firstly, although the introduction highlights the potential of radiomics, providing more context on previous studies utilizing radiomics in pediatric brain tumors would offer additional background for readers unfamiliar with the field. Secondly, the inclusion of more direct visual aids, such as feature importance graphs, would help readers better understand the key findings. Additionally, placing greater emphasis on the potential clinical implications and how the results could be translated into practice would be beneficial.

Finally, there are two other elements to consider: the sample size of 70 subjects, while respectable, is relatively small for a study involving multiple subgroups and extensive feature extraction. This could limit the statistical power of the results and their generalizability. Furthermore, although the multi-institutional nature of the data is a strength, the study lacks an external validation cohort. Including such a cohort would strengthen the conclusions and provide a more rigorous test of the model's generalizability. However, I believe these points can be addressed in subsequent phases of the study, given that this is a preliminary investigation.

Overall, I consider it an excellent and certainly innovative work. I therefore believe it deserves publication once the authors have addressed the aforementioned elements.

Author Response

A radiomic approach for evaluating intra-subgroup heterogeneity in SHH and Group 4 pediatric medulloblastoma: A preliminary multi-institutional study

We thank the editor and the reviewers for their feedback on our manuscript. We appreciate the enthusiastic comments of the reviewers on the merits of our work such as “well-written”, “robust statistical analysis”, and “excellent and certainly innovative work”. Below, we provide the point-by-point response to the reviewer’s comments. The reviewers’ comments are reproduced in italicized Times New Roman blue font and our responses are provided in black Times New Roman font. Please note that the references in the responses are included in (Author Surname, Year Published) citation style, to distinguish them from the ones in the revised manuscript. All the changes have been highlighted in the revised manuscript. 

Reviewer #1:

  1. Although the introduction highlights the potential of radiomics, providing more context on previous studies utilizing radiomics in pediatric brain tumors would offer additional background for readers unfamiliar with the field.

We thank the reviewer for the comment. We have now elaborated on some of the previous works that employed radiomic approaches in the introduction section and highlighted some of the frequent techniques employed in the context of pediatric brain tumors. Specifically, we mention:

 “For instance, the studies in [Grist et al., 2021, Iyer et al., 2022, Liu at al., 2021, Yan et al., 2020, Zheng et al., 2022] attempted to employ statistical techniques, such as univariate and multivariate logistic regression, to predict survival in MB. Regression models (e.g., LASSO) have been incorporated in these works, followed by Kaplan Meier estimate for risk-stratification. Similarly, the studies in [Iyer at al., 2022, Chang et al., 2021, Iv et al., 2019, Saju et al., 2022, Wang et al., 2023, Yan et al., 2020, Zhang et al., 2022, Dasgupta et al., 2019] employed several machine learning classifiers (e.g., SVM) along with some statistical techniques, such as logistic regression and ANOVA, to predict the 4 molecular subgroups in MB.”

  1. The inclusion of more direct visual aids, such as feature importance graphs, would help readers better understand the key findings.

We thank the reviewer for the suggestion. We have now included feature importance graphs for the 2 subgroups in the supplementary material, representing shape features, texture features, and both feature families combined into our descriptor, which we renamed to  mRRisk (medulloblastoma radiomics risk), to distinguish it as a feature descriptor and not a risk score. In Figures 1-3 below, we display the importance graphs for the shape, texture, and mRRisk descriptor features, respectively, for each of SHH and Group 4 subgroups.  The x-axis represents the F-score, with a higher score indicating that the feature has larger effect on risk-stratification.

Figure 1: Feature importance graphs for shape features for both SHH and Group 4 subgroups. The y-axis represents the feature number whereas the x-axis represents the F-score for each feature.

Figure 2: Feature importance graphs for texture features for both SHH and Group 4 subgroups. The y-axis represents the feature number whereas the x-axis represents the F-score for each feature.

 Figure 3: Feature importance graphs for mRRisk descriptor features for both SHH and Group 4 subgroups. The y-axis represents the feature number whereas the x-axis represents the F-score for each feature.

  1. Placing greater emphasis on the potential clinical implications and how the results could be translated into practice would be beneficial.

We thank the reviewer for the suggestion. We have now included the following text in the Discussion section, to emphasize the clinical implications of our proposed prognostic signature and how it can be translated into clinical practice:

“While there are many radiomic approaches in the literature that aimed at molecular subgroup classification [Ismail et al., 2023], our study uniquely seeks to utilize those radiomic tools to further sub-stratify patients within the individual subgroups. Further optimization of mRRisk with rigorous validation on large multi-institutional cohorts would allow for enriched risk-stratification. Specifically, our prognostic signature can help reduce the long-term toxicities within the children with MB that are recognized as average-risk by providing additional prognostic insights to tailor their treatment intensity, Additionally, the signature can also enable identification of patients that are true candidates for therapy intensification.  This can lead to incorporating the signature in clinical trials that aim for therapy de-escalation as well as therapy intensification, which could improve patients’ outcomes and treatment planning”.

  1. There are two other elements to consider: the sample size of 70 subjects, while respectable, is relatively small for a study involving multiple subgroups and extensive feature extraction. This could limit the statistical power of the results and their generalizability.

Furthermore, although the multi-institutional nature of the data is a strength, the study lacks an external validation cohort. Including such a cohort would strengthen the conclusions and provide a more rigorous test of the model's generalizability. However, I believe these points can be addressed in subsequent phases of the study, given that this is a preliminary investigation.

We thank the reviewer for the excellent comments and suggestions. We have acknowledged in limitations, the fact that our sample size is small and that the statistical power is limited.  Our goal is to continue curating multi-institutional MB studies with known molecular subgroup information so we can validate our approaches on larger cohorts. We have requested access to the scans from the ACNS0331, which is a completed clinical trial in medulloblastoma patients for therapy de-escalation, where subjects were all uniformly treated and with known molecular subgroup characteristics. This will allow us to conduct survival analysis on larger cohorts to possibly sub-stratify patients with similar molecular characteristics.

References

  1. Grist, J.T.; Withey, S.; Bennett, C.; Rose, H.E.; MacPherson, L.; Oates, A.; Powell, S.; Novak, J.; Abernethy, L.; Pizer, B.; et al. Combining multi-site magnetic resonance imaging with machine learning predicts survival in pediatric brain tumors. Sci. Rep. 2021, 11, 18897.
  2. Yan, J.; Zhang, S.; Li, K.K.; Wang, W.; Li, K.; Duan, W.; Yuan, B.; Wang, L.; Liu, L.; Zhan, Y.; et al. Incremental prognostic value and underlying biological pathways of radiomics patterns in medulloblastoma. eBioMedicine 2020, 61.
  3. Liu, Z.M.; Zhang, H.; Ge, M.; Hao, X.L.; An, X.; Tian, Y.J. Radiomics signature for the prediction of progression-free survival and radiotherapeutic benefits in pediatric medulloblastoma. Child’s Nerv. Syst. 2022, 38, 1085–1094.
  4. Zheng, H.; Li, J.; Liu, H.; Ting, G.; Yin, Q.; Li, R.; Liu, M.; Zhang, Y.; Duan, S.; Li, Y.; et al. MRI Radiomics Signature of Pediatric Medulloblastoma Improves Risk Stratification Beyond Clinical and Conventional MR Imaging Features. J. Magn. Reson. Imaging 2022, 58, 236–246.
  5. Iyer, S.; Ismail, M.; Tamrazi, B.; Salloum, R.; de Blank, P.; Margol, A.; Correa, R.; Chen, J.; Bera, K.; Statsevych, V.; et al. Novel MRI deformation-heterogeneity radiomic features are associated with molecular subgroups and overall survival in pediatric medulloblastoma: Preliminary findings from a multi-institutional study. Front. Oncol. 2022, 12, 915143.
  6. Chang, F.C.; Wong, T.T.; Wu, K.S.; Lu, C.F.; Weng, T.W.; Liang, M.L.; Wu, C.C.; Guo, W.Y.; Chen, C.Y.; Hsieh, K.L. Magnetic resonance radiomics features and prognosticators in different molecular subtypes of pediatric Medulloblastoma. PLoS ONE 2021, 16, e0255500.
  7. Iv, M.; Zhou, M.; Shpanskaya, K.; Perreault, S.; Wang, Z.; Tranvinh, E.; Lanzman, B.; Vajapeyam, S.; Vitanza, N.A.; Fisher, P.G.; et al. MR imaging–based radiomic signatures of distinct molecular subgroups of medulloblastoma. Am. J. Neuroradiol. 2019, 40, 154–161.
  8. Saju, A.C.; Chatterjee, A.; Sahu, A.; Gupta, T.; Krishnatry, R.; Mokal, S.; Sahay, A.; Epari, S.; Prasad, M.; Chinnaswamy, G.; et al. Machine-learning approach to predict molecular subgroups of medulloblastoma using multiparametric MRI-based tumor radiomics. Br. J. Radiol. 2022, 95, 20211359.
  9. Wang, Y.; Wang, L.; Qin, B.; Hu, X.; Xiao, W.; Tong, Z.; Li, S.; Jing, Y.; Li, L.; Zhang, Y. Preoperative prediction of sonic hedgehog and group 4 molecular subtypes of pediatric medulloblastoma based on radiomics of multiparametric MRI combined with clinical parameters. Front. Neurosci. 2023, 17, 1157858.
  10. Yan, J.; Liu, L.; Wang, W.; Zhao, Y.; Li, K.K.; Li, K.; Wang, L.; Yuan, B.; Geng, H.; Zhang, S.; et al. Radiomic features from multi-parameter MRI combined with clinical parameters predict molecular subgroups in patients with medulloblastoma. Front. Oncol. 2020, 10, 558162.
  11. Zhang, M.; Wong, S.W.; Wright, J.N.; Wagner, M.W.; Toescu, S.; Han, M.; Tam, L.T.; Zhou, Q.; Ahmadian, S.S.; Shpanskaya, K.; et al. MRI radiogenomics of pediatric medulloblastoma: A multicenter study. Radiology 2022, 304, 406–416.
  12. Dasgupta, A.; Gupta, T.; Pungavkar, S.; Shirsat, N.; Epari, S.; Chinnaswamy, G.; Mahajan, A.; Janu, A.; Moiyadi, A.; Kannan, S.; et al. Nomograms based on preoperative multiparametric magnetic resonance imaging for prediction of molecular subgrouping in medulloblastoma: Results from a radiogenomics study of 111 patients. Neuro-Oncology 2019, 21, 115–124.
  13. Ismail, M., Craig, S., Ahmed, R., de Blank, P. and Tiwari, P., 2023. Opportunities and Advances in Radiomics and Radiogenomics for Pediatric Medulloblastoma Tumors. Diagnostics, 13(17), p.2727.

Reviewer 2 Report

Comments and Suggestions for Authors

In this manuscript, Ismail et al. presented a radiomic approach that could be used to evaluate the prognosis of medulloblastoma. Overall, the study was well conceptualized. However, there are a few concerns regarding the methodology that the authors have used in this research.

1.    The authors may want to clarify the methods that they use to construct mRRisc. The current overview as listed in the 2.1 is not perspicuous. To be specific, the authors may want to explain the definition for texture and shape features. How are those features applied to the algorithm that would predict the survival risk score.

2.    For method 2.3, the authors may want to provide the details of the 214 texture features as well as 18 morphological features in the supplementary files. A work flow chart could be helpful at this point the walk the readers through the data extraction process.

3.    The authors may want to include the KM curve for Group 4 when employed mRRisc. In addition, please check the font size and the resolution of all the figures. The font size should be consistent with each other (Figure 2e compared with other figures in the same panel), and the resolution should be improved.

Comments on the Quality of English Language

Overall the English Language is fine. Minor polish may be needed when the revision is done. 

Author Response

A radiomic approach for evaluating intra-subgroup heterogeneity in SHH and Group 4 pediatric medulloblastoma: A preliminary multi-institutional study

We thank the editor and the reviewers for their feedback on our manuscript. We appreciate the enthusiastic comments of the reviewers on the merits of our work such as “well conceptualized. Below, we provide the point-by-point response to the reviewer’s comments. The reviewers’ comments are reproduced in italicized Times New Roman blue font and our responses are provided in black Times New Roman font. Please note that the references in the responses are included in (Author Surname, Year Published) citation style, to distinguish them from the ones in the revised manuscript. All the changes have been highlighted in the revised manuscript. 

Reviewer #2:

  1. The authors may want to clarify the methods that they use to construct mRRisc. The current overview as listed in the 2.1 is not perspicuous. To be specific, the authors may want to explain the definition for texture and shape features. How are those features applied to the algorithm that would predict the survival risk score.

We thank the reviewer for the comment. We have now renamed our signature to mRRisk (medulloblastoma radiomics risk), to distinguish it as a feature descriptor and not a risk score.

Our signature, mRRisk, is constructed by concatenating the 2 feature families we extracted from the tumor regions (texture and shape (both explained in detail in Subsection 2.3)) into one vector. After feature concatenation, statistical methods were applied on mRRisk, particularly, multivariate logistic regression (including Elastic Net, LASSO regression, and ridge regression) to pick the important features and remove the ones that are not contributing to risk-stratification. We have now elaborated on the details mentioned in Section 2.1 to provide a more comprehensive overview of the approach, while more details are provided in Subsections 2.2 - 2.4. Specifically, we mention in Subsection 2.1:

Following preprocessing and tumor segmentation, we extract radiomic features that quantify both the morphological and textural attributes of the pediatric MB tumors. Following feature extraction, our prognostic signature, mRRisk, is constructed by concatenating the 2 feature families into one vector. Statistical methods were then applied on mRRisk, particularly, multivariate logistic regression (including Elastic Net, LASSO regression, and ridge regression) for feature pruning and selecting the features that are statistically significant in risk-stratification while shrinking those that do not contribute to risk assessment. The top features were then employed to create a continuous survival risk score that stratifies all the patients into low- and high-risk groups, within each molecular subgroup (SHH, Group 4).

  1. For method 2.3, the authors may want to provide the details of the 214 texture features as well as 18 morphological features in the supplementary files. A work flow chart could be helpful at this point the walk the readers through the data extraction process.

We thank the reviewer for the comment. We have now provided the names of the features in the supplementary material and provided detailed information in the Methods section on the 2 feature families we have (shape and texture). Specifically, we mention:

Namely, the texture features [Ghalati et al., 2021] encompass gradient, Haralick, Laws, Gabor, and COLLAGE (gradient entropy) features [Prasanna et al., 2016] and are computed for every voxel of every tumor region. We utilize these per-voxel measurements to compute first order statistics (mean, median, standard deviation, skewness, and kurtosis) per feature for every tumor region. This resulted in 1070 textural features for every tumor region, as well as for the tumor habitat. Additionally, 18 morphological features were extracted from the different tumor regions. Namely, 4 local features that capture surface-based irregularities (Curvedness, Sharpness, Shape Index, and Total Curvature) were computed from each region. The 4 local features were computed from the constructed isosurfaces of the tumor regions, followed by computing the first and second fundamental forms of the surfaces [Pineaar et al, 2008]. Gaussian and mean curvatures were then computed from the fundamental forms per voxel. Finally, the local features were derived from the Gaussian and mean curvatures.

  1. The authors may want to include the KM curve for Group 4 when employed mRRisk. In addition, please check the font size and the resolution of all the figures. The font size should be consistent with each other (Figure 2e compared with other figures in the same panel), and the resolution should be improved.

We thank the reviewer for the comment and the suggestion. We have now fixed the size and resolution for Figure 2 and have also included the KM curves when mRRisk we applied on Group 4 subgroup for risk-stratification. These curves are now included in a separate figure (Figure 3).

References:

  1. Ghalati, M.K., Nunes, A., Ferreira, H., Serranho, P. and Bernardes, R., 2021. Texture analysis and its applications in biomedical imaging: A survey. IEEE Reviews in Biomedical Engineering, 15, pp.222-246.
  2. Prasanna, P., Tiwari, P. and Madabhushi, A., 2016. Co-occurrence of local anisotropic gradient orientations (CoLlAGe): a new radiomics descriptor. Scientific reports, 6(1), p.37241.
  3. Pienaar, R., Fischl, B., Caviness, V., Makris, N. and Grant, P.E., 2008. A methodology for analyzing curvature in the developing brain from preterm to adult. International journal of imaging systems and technology, 18(1), pp.42-68.
